# Radiomics-Based Machine Learning for Predicting the Injury Time of Rib Fractures in Gemstone Spectral Imaging Scans

**DOI:** 10.3390/bioengineering10010008

**Published:** 2022-12-21

**Authors:** Liang Jin, Yingli Sun, Zongjing Ma, Ming Li

**Affiliations:** 1Shanghai Key Lab of Forensic Medicine, Key Lab of Forensic Science, Ministry of Justice, Academy of Forensic Science, Shanghai 200063, China; 2Radiology Department, Huadong Hospital, Affiliated with Fudan University, Shanghai 200040, China; 3Institute of Functional and Molecular Medical Imaging, Shanghai 200040, China

**Keywords:** chest trauma, human-model collaboration, injury time, radiomics-based model, rib fractures

## Abstract

This retrospective study aimed to predict the injury time of rib fractures in distinguishing fresh (30 days) or old (90 days) rib fractures. We enrolled 111 patients with chest trauma who had been scanned for rib fractures at our hospital between January 2018 and December 2018 using gemstone spectral imaging (GSI). The volume of interest of each broken end of the rib fractures was segmented using calcium-based material decomposition images derived from the GSI scans. The training and testing sets were randomly assigned in a 7:3 ratio. All cases were divided into groups distinguishing the injury time at 30 and 90 days. We constructed radiomics-based models to predict the injury time of rib fractures. The model performance was assessed by the area under the curve (AUC) obtained by the receiver operating characteristic analysis. We included 54 patients with 259 rib fracture segmentations (34 men; mean age, 52 years ± 12.02; and range, 19–72 years). Nine features were excluded by the least absolute shrinkage and selection operator logistic regression to build the radiomics signature. For distinguishing the injury time at 30 days, the Support Vector Machine (SVM) model and human–model collaboration resulted in an accuracy and AUC of 0.85 and 0.871 and 0.91 and 0.912, respectively, and 0.81 and 0.804 and 0.83 and 0.85, respectively, at 90 days in the testing set. The radiomics-based model displayed good accuracy in differentiating between the injury time of rib fractures at 30 and 90 days, and the human–model collaboration generated more accurate outcomes, which may help to add value to clinical practice and distinguish artificial injury in forensic medicine.

## 1. Introduction

Rib fractures play an important role in the diagnosis and treatment of patients with blunt chest trauma [1,2]. Complications associated with rib fractures can exert a substantial effect on morbidity and mortality, the number of which is estimated to be up to 12% of patients with traumatic rib fractures [2,3]. Computed tomography (CT) image features of rib fractures can consist of a sharp fracture line without periosteal reaction or callus formation (fresh fractures, within 3 weeks) [4], blurred edges of the fracture line with callus formation (union fractures) [5], or mature callus, bone remodeling, and invisible features of the fracture line (old fracture, 12 weeks). However, the time of callus growth following fracture varies with the age; it is not possible to accurately estimate the injury time of rib fractures based on routine CT image features [1]. Accurate detection of rib fractures with the injury time will add value to the clinical practice. Moreover, the injury time of rib fractures will facilitate distinguishing artificial injuries in forensic medicine [6]. 

Dual-energy CT gemstone spectral imaging (GSI) can differentiate and classify materials through the fast kilovoltage-switching technique [7,8,9,10,11], thereby creating accurate material decomposition images (e.g., calcium- and iodine-based material decomposition images) [9]. Following fractures, the periosteum and endosteum near the broken end begin to proliferate and thicken. In addition, blood vessels invade and form bone by intramembranous ossification; contrarily, the majority of fibrovascular granulation tissues formed by the organization of the hematoma are transformed into the cartilage between the fractured ends and under the lifted periosteum. Endochondral ossification leads to bone formation. Theoretically, the bone mineral content of the broken end is different at any time following a fracture. The calcium-based material decomposition images from GSI can be used to measure bone mineral content. However, there are no previous studies for reference and no standard for measurement.

Recently, radiomics has gained popularity in medical imaging worldwide. It is generally applied to extract quantitative and ideally reproducible information from diagnostic images, including complex patterns that are difficult to recognize or quantify by the human eye [12,13]. Hence, we hypothesized that radiomics analysis could extract the segmented volume of interest (VOI) of each broken end of rib fractures in GSI scans to predict the possible injury time of ribs to distinguish the injury time of rib fractures within 30 days (approximately 4 weeks) or 90 days (approximately 12 weeks); that is, the aim of this study was to predict the injury time of rib fracture in distinguishing fresh (30 days) or old (90 days) rib fractures.

## 2. Materials and Methods

### 2.1. Radiomics Workflow 

The radiomics workflow is presented in Figure 1, including (1) the image collection, (2) lesion segmentation and radiomic feature extraction, (3) feature selection and model construction, and (4) prediction performance evaluation. 

### 2.2. Patients’ Enrollment and Image Collection

The Institutional Review Board of our hospital approved this retrospective study and waived the need for informed consent (code: 20220051). We enrolled 111 patients with chest trauma scanned for rib fracture evaluation between January 2018 and December 2018 using GSI scanning. The inclusion criteria were as follows: (1) patients with trauma and thin-slice chest-abdomen CT images (1–1.25 mm) containing all ribs, (2) without surgical internal fixation for rib fractures, and (3) with thin-slice CT images without a breathing artifact debasing the diagnostic accuracy. The exclusion criterion was as follows: the data file of GSI scanning for image post-processing is missing.

GSI scanning was performed using a 16 cm wide-coverage detector CT scanner (Revolution CT, GE Healthcare, Milwaukee, WI, USA), with a fast tube voltage switching between 80 kVp and 140 kVp, including all ribs. We used the following spectral imaging acquisition protocol for chest-abdomen CT: 200 mAs; helical scan with pitch, 0.984:1; rotation time, 0.5 s; collimation thickness, 0.625 mm; and the reconstruction field of view, 50 cm. Based on the acquired raw imaging data, the calcium-based material decomposition images were used for further segmentation. 

Owing to the particularity and balance of the data, the time of fracture occurrence of the patients in this dataset was >30 days and >90 days as the classification criteria. The testing dataset and training dataset were separated by a random method based on a 3:7 ratio, and the random seeds were 32.

### 2.3. Image Segmentation

Following CT examinations, all enrolled patients were randomly diagnosed by two radiologists, A and B, who did not participate in this study, in the radiology department for 48 h. The diagnosed CT reports were considered the reference standards for further VOI segmentations. All GSI scans were sent to an advantage workstation 4.7 for further image postprocessing. First, all images were analyzed using the material decomposition function in the GSI General software, which transferred the images to calcium-based material decomposition images (Figure 1). Subsequently, radiologist C with 8 years of experience in chest diagnosis manually delineated the VOI of the broken ends of the fractured ribs by referring to the diagnosed CT reports (by radiologists A and B). A medical software ITK-SNAP (Version 3.6.0) was used for the segmentation. 

### 2.4. Radiologist’s Interpretation for the Evaluation of the Injury Time of Rib Fractures 

For each patient, the time between the injury date and examination date was identified as the accurate injury time of rib fractures for the reference standard. A chest radiologist with 12 years of experience (radiologist D) in rib fracture diagnosis, blinded to any information regarding the injury time of rib fracture analysis, interpreted all CT assessments with both routine CT images and calcium-based material decomposition images. For all enrolled patients, radiologist D interpreted if the injury time was within 30 days in the initial round or within 90 days in the subsequent round. 

### 2.5. Feature Extraction and Selection

The imaging features were automatically extracted from the CT images with the Radcloud platform (http://radcloud.cn/ accessed on 30 January 2022). These features were categorized into four groups as follows: (i) Group 1 (first-order statistics) consisted of 18 descriptors that quantitatively delineated the distribution of VOL intensities within the MR image through commonly used and basic metrics; (ii) Group 2 (shape- and size-based features) contained 14 three-dimensional features that reflected the shape and size of the region; (iii) Group 3 (texture features) comprised 75 textural features that could quantify the region heterogeneity differences, calculated from the Gray Level Co-occurrence Matrix, Gray Level Run Length Matrix, Gray Level Size Zone Matrix, Gray Level Difference Matrix, and Neighborhood Gray-Tone Difference Matrices; and (iv) Group 4 (higher order statistics features) included the intensity and texture features derived from filter transformation of the original image. We used seven types of filters as follows: exponential, square, square root, logarithm, gradient, lbp-2D, and wavelet (wavelet-LLL, wavelet-HHH, wavelet-HLL, wavelet-HHL, wavelet-LLH, wavelet-HLH, wavelet-LHL, and wavelet-LHH). 

As described above, numerous image features may be computed. However, these extracted features may not be useful for a particular task. This warrants dimensionality reduction and the selection of task-specific features for best performance. To reduce the redundant features, the feature selection methods included the variance threshold (variance threshold= 0.8). We used the Select K Best and the least absolute shrinkage and selection operator (LASSO) for this purpose. The threshold was 0.8 for the variance threshold method, such that the eigenvalues were removed for variances <0.8. The Select K Best method, which belongs to a single variable feature selection method, used *p*-values to analyze the relationship between the features and classification results; all features with *p*-value < 0.05 were used. This method is intended to minimize the LASSO cost function and to obtain all features with non-zero coefficients. The minimized objective function was as follows: $$\underset{w}{\min\frac{1}{2n}}||Xw-y|{|}_{2}^{2}+\alpha ||w|{|}_{1}$$
where *X* is a matrix of the radiomic features, *y* is a vector of sample labels, *n* is the number of samples, *w* is a coefficient vector of the regression model, and $\alpha ||w|{|}_{1}$ is the LASSO penalty with the constant α and the ℓ1-norm of the coefficient vector $||w|{|}_{1}$. In the LASSO model, the L1 regularizer was used as the cost function. The error value of cross-validation was 5, and the maximum number of iterations was 1000.

### 2.6. Performance of the Radiomics Signature and Statistical Analysis 

We performed the statistical analysis in the Radcloud platform. Based on the selected features, several supervised learning models are available for classification analysis, which creates models that attempt to separate or predict the data with respect to an outcome or phenotype (for instance, patient outcomes or response). In this study, the radiomics-based models were constructed with four machine learning models, namely Support Vector Machine (SVM), Random Forest (RF), Logistic Regression (LR), and Gaussian NB (GNB), and the validation method was used to improve the effectiveness of the model. SVM contained the following parameters: kernel (rbf), C (1), gamma (auto), class_weight (balanced), and decision_function_shape (ovr). LR contained the following parameters: penalty (L2), C (1), solver (liblinear), class_weight (None), and multi_class (ovr). GNB consisted of the following parameters: priors (None) and var-smoothing = 1 × 10^−9^.

To assess the predictive performance of the model, the receiver operating characteristic (ROC) curve, namely area under the curve (AUC), was used both in the training dataset and validation dataset, respectively. Moreover, indicators including P (precision = true positives/(true positives+ false positives)), R (recall = true positives/(true positives+ false negatives)), f1-score (f1-score = P×R×2/(P + R)), TP (positive samples predicted by the model as positive classes), TN (negative samples predicted by the model as a negative class), FP (negative samples predicted to be positive by the model), and FN (positive samples predicted to be negative by the model) were used to support (total number in the test set) and to evaluate the performance of the model in this study. Macro-average = (P1 + P2)/2, weighted average = P1×(TP/TP + FP) +P2×(FP/TP + FP).

## 3. Results

### 3.1. Patient Characteristics

Between January 2018 and December 2018, 111 patients underwent GSI scanning. There were 42 and 15 patients with surgical internal fixation for rib fractures and a breathing artifact debasing diagnostic accuracy, respectively. We included 54 patients with 295 rib fracture segmentations (34 men and 20 women; mean age, 52 years ± 12.02; and range, 19–72 years). The average dose length product of all the patients was 503.01 mGy*cm, and the average volume CT dose index was 10.41 mGy. Of the 54 patients, the injury time of rib fractures was <30 days in 9 (16.7%) patients. However, the injury time was <90 days in twenty-seven (50%) patients (Figure 2).

### 3.2. Feature Extraction and Selection

A total of 1,409 radiomics features were extracted. We selected the 90 most valuable features after using Select K Best. Eventually, the LASSO logistic regression excluded nine features to build the radiomics signature. Figure 1 and Table 1 depict the optimal parameter λ(0.078) of each fold and the selected features of the corresponding fold. Figure 1 depicts the histogram of the Rad-score. 

### 3.3. Prediction Performance of the Radiomics-Based Models and Radiologists

#### 3.3.1. Prediction Performance on Distinguishing 30 Days of the Injury Time of Rib Fractures 

The SVM learning model achieved the best classification effect. Appendix A summarizes the performance of other machine learning models (RF, LR, and GNB). We used the precision, recall, and f1-score to evaluate the performance of the SVM model, radiologist’s interpretation, and human–model collaboration. The SVM model, radiologist’s interpretation, and human–model collaboration resulted in an accuracy and AUC of 0.85 and 0.871, 0.74 and 0.69, and 0.91 and 0.912, respectively, in the testing set (Table 2; Figure 3). The final model had a Brier score of 0.082. Figure 4 depicts its calibration curve in the testing set.

#### 3.3.2. Prediction Performance on Distinguishing 90 Days of the Injury Time of Rib Fractures 

The SVM model, radiologist’s interpretation, and human–model collaboration resulted in an accuracy and AUC of 0.81 and 0.804, 0.71 and 0.667, and 0.83 and 0.85, respectively, in the testing set (Table 3; Figure 5). The final model had a Brier score of 0.09. Figure 4 depicts its calibration curve in the testing set. 

## 4. Discussion

The present study aimed to predict the injury time of rib fractures based on radiomic features derived from the segmented VOI of each broken end of rib fractures in GSI scans. Our proposed SVM model showed a better prediction of the injury time of rib fractures by distinguishing 30 days and 90 days. The model displayed better performance than the experienced radiologist’s interpretation for both time points, thus indicating that the radiomics-based model could assist radiologists in evaluating the injury time of rib fractures with satisfactory performance.

Artificial intelligence techniques, including deep learning, radiomics, and radiomics-based machine learning, are valuable for clinical diagnosis and treatment in medical practice [14,15,16,17,18,19]. The majority of previous studies have focused on the detection and segmentation of rib fractures using deep learning [17,20]; nonetheless, few studies have attempted to classify rib fractures [20,21] and demonstrated that artificial intelligence could significantly reduce the workload of doctors while improving diagnostic accuracy. Machine learning focused on how computers learn from data [22] could provide some additional medical information from imaging which is hard for human eyes to explain in medical practice; patients may benefit from this medical information based on machine learning approaches. Ji et al. used machine-learning analysis for contrast-enhanced CT radiomics to predict the recurrence of hepatocellular carcinoma after resection [23]; Draelos et al. used machine-learning models to predict multiple abnormalities with large-scale chest CT volumes [24]. However, no reports have confirmed the injury time of rib fractures using such machine-learning techniques. 

Rib fracture healing is affected by age [1,2], and evaluating the injury time of rib fractures from the imaging features of rib fractures is difficult even for an experienced radiologist (Figure 6), despite the existing knowledge (including morphological features) and experience. Our findings indicated and confirmed this phenomenon. First, the nine selected features (Table 1) correlated significantly with the grayscale. Machine learning showed better performance in distinguishing 30 days or 90 days (accuracy: 0.85 and 0.81, respectively) compared to the radiologists’ interpretation (0.74 and 0.71, respectively). Our results indicated machine learning could learn some features to help predict the injury time of rib fractures. Moreover, our results showed a better performance in distinguishing 30 days than 90 days in both machine learning and the radiologists’ interpretation; this may be explained by the natural characteristic of fracture healing. The less confusing imaging features of fracture healing within 30 days may make it easier to predict the injury time. However, an unsupervised radiomics-based model may not be superior to human performance despite the higher accuracy. Radiologists’ interpretation can be assisted by the prediction performance of the model; the increased accuracy of distinguishing 30 days and 90 days displayed effective complementarity following human–model collaboration, consistent with the results of previous artificial intelligence-based studies [17,19]. In the present study, we set the injury time of fresh fractures within 4 weeks (30 days). This could be attributed to the absence of an accepted clinical definition of fresh fractures [4] and the influence of age. Moreover, we set the injury time at 12 weeks (90 days) owing to the consensus of old fractures. For the rib fractures at 30 days, the imaging features principally comprised a sharp fracture line without periosteal reaction or callus formation, which was easily captured by the model or detected by human eyes. However, a comparison of the features up to 90 days, such as blurred edges of the fracture line with callus formation, mature callus, bone remodeling, and invisible features of the fracture line, added to the confounding factors for the evaluation. This finding may explain the better prediction of the injury time of rib fractures within 30 days. In the present study, we summarized the performance of other machine learning models (RF, LR, and GNB) in Appendix A. Some models showed better performance than SVM in terms of overfitting, some models did not show satisfactory performance compared to SVM, and, finally, we chose SVM as the best model. 

The present study had several limitations. Researchers have established GSI scans as a novel tool to provide additional information for clinical decision-making [7,8,9,10,11]; thus, we used GSI data for rib fracture segmentation. However, we did not compare the performance of radiomics prediction between the conventional CT images and GSI scans with different image modalities; our findings could not indicate whether GSI scans could contribute more than conventional CT images, and this needs to be addressed in future studies. Nevertheless, our radiomics prediction already demonstrated better performance both by the model and human–model collaboration. Second, we performed a single-center study without external validation datasets; thus, determining the model’s robustness was challenging, and this will be addressed in future studies.

## 5. Conclusions

The imaging features derived from the GSI scan-based radiomics model displayed good accuracy in differentiating the injury time of rib fractures within 30 and 90 days, and the human–model collaboration demonstrated more accurate outcomes. Our novel study used a radiomics-based model to predict the injury time of rib fractures, demonstrated better performance, and is expected to be valuable in clinical and forensic decision-making.

## Figures and Tables

**Figure 1 bioengineering-10-00008-f001:**
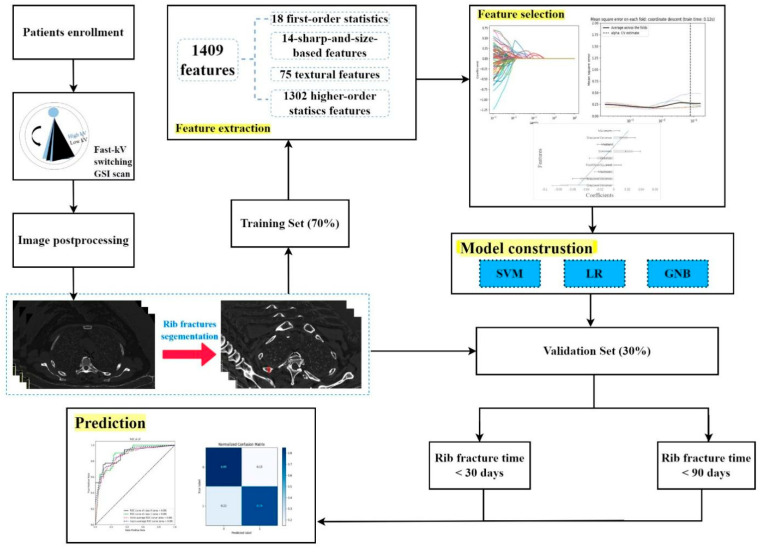
Radiomics workflow.

**Figure 2 bioengineering-10-00008-f002:**
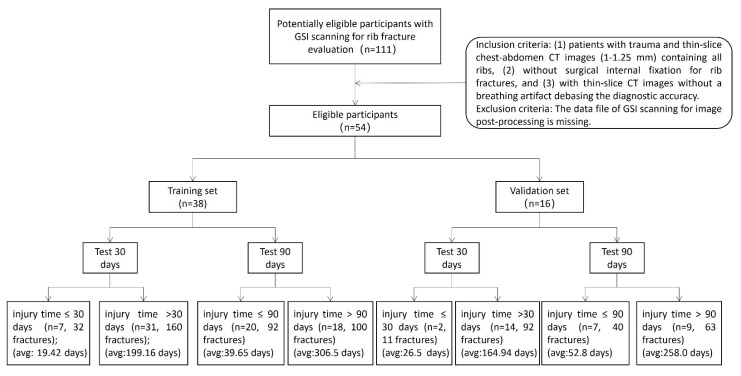
The patients’ enrollment.

**Figure 3 bioengineering-10-00008-f003:**
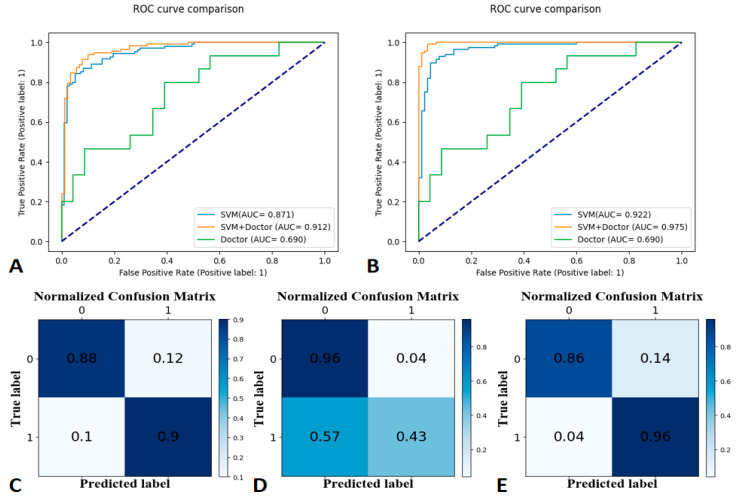
Receiver operating characteristic (ROC) curves for the training set at 30-day labels (**A**), ROC curves for the testing set at 30-day labels (**B**), a confusion matrix of the Support Vector Machine (**C**), human–model collaboration (**D**), and radiologist’s interpretation (**E**) for the testing set (30-day label).

**Figure 4 bioengineering-10-00008-f004:**
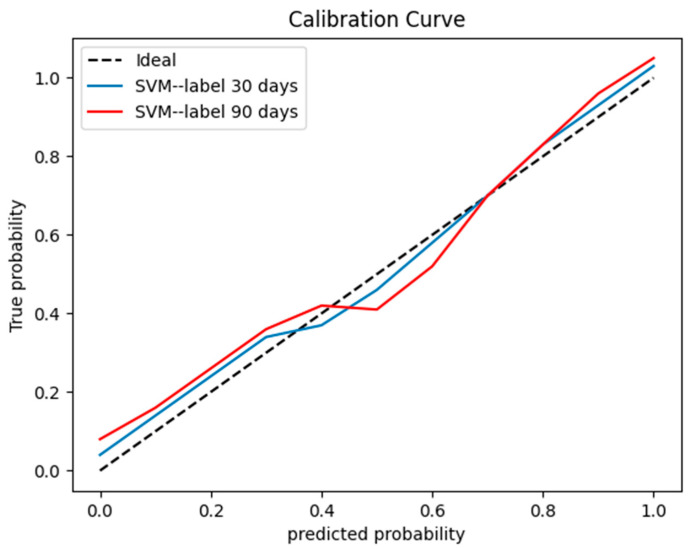
A calibration curve for the testing set.

**Figure 5 bioengineering-10-00008-f005:**
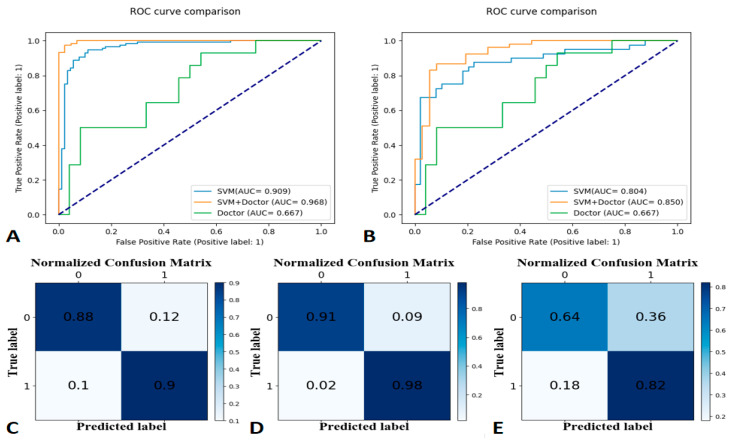
Receiver operating characteristic (ROC) curves for the training set at 90-day labels (**A**), ROC curves for the testing set at 90-day labels (**B**), a confusion matrix of the Support Vector Machine (**C**), human–model collaboration (**D**), and radiologist’s interpretation (**E**) for the testing set (90-day label).

**Figure 6 bioengineering-10-00008-f006:**
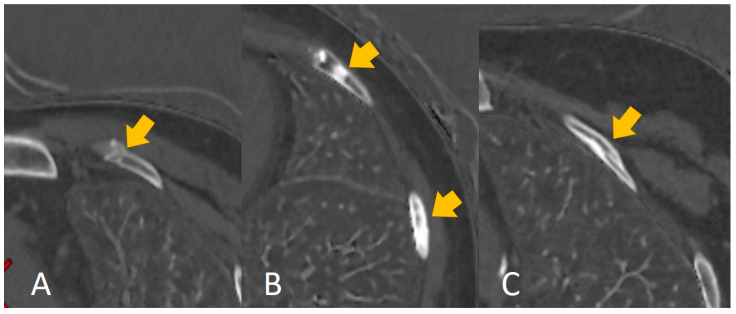
The calcium-based material decomposition images: (**A**) shows the injury time of a rib fracture within 30 days; (**B**) shows the injury time of a rib fracture between 30 and 90 days, and (**C**) shows the injury time of a rib fracture over 90 days.

**Table 1 bioengineering-10-00008-t001:** A description of the selected radiomic features with their associated feature group and filters.

Radiomic Feature	Radiomic Class	Filter
GrayLevelVariance	glszm	original
GrayLevelVariance	glszm	logarithm
Maximum	first-order	gradient
RootMeanSquared	first-order	gradient
Variance	first-order	gradient
Skewness	first-order	squareroot
Median	first-order	squareroot
GrayLevelVariance	glszm	squareroot
Maximum	first-order	wavelet-HHL

Label: GLSZM = Gray Level Size Zone Matrix.

**Table 2 bioengineering-10-00008-t002:** Performances of the radiomics-based model, radiologist’s interpretation, and human–model collaboration for the testing set in predicting rib fracture at 30 days.

Fracture Time < 30 Days	Testing Set (30%)		
	Precision	Recall	F1-Score
SVM model			
Positive	0.82	0.94	0.88
Negative	0.91	0.75	0.82
Macro average	0.87	0.84	0.85
Weighted average	0.86	0.85	0.85
Accuracy	0.85		
AUC	0.871		
Radiologist’s interpretation			
Positive	0.78	0.47	0.58
Negative	0.72	0.91	0.81
Macro average	0.75	0.69	0.70
Weighted average	0.75	0.74	0.72
Accuracy	0.74		
AUC	0.69		
Human–model collaboration			
Positive	0.97	0.83	0.89
Negative	0.88	0.98	0.93
Macro average	0.93	0.91	0.91
Weighted average	0.92	0.91	0.91
Accuracy	0.91		
AUC	0.912		

SVM, Support Vector Machine; AUC, area under the receiver operating characteristic curve.

**Table 3 bioengineering-10-00008-t003:** Performances of the radiomics-based model, radiologist’s interpretation, and human–model collaboration for the testing set in predicting rib fracture at 90 days.

Fracture Time < 90 Days	Testing Set (30%)		
	Precision	Recall	F1-Score
SVM model			
Positive	0.81	0.75	0.78
Negative	0.81	0.86	0.83
Macro average	0.81	0.80	0.81
Weighted average	0.81	0.81	0.81
Accuracy	0.81		
AUC	0.804		
Radiologist’s interpretation			
Positive	0.64	0.50	0.78
Negative	0.74	0.83	0.56
Macro average	0.69	0.79	0.67
Weighted average	0.70	0.79	0.70
Accuracy	0.71		
AUC	0.667		
Human–model collaboration			
Positive	0.95	0.75	0.82
Negative	0.72	0.94	0.84
Macro average	0.84	0.85	0.83
Weighted average	0.86	0.83	0.83
Accuracy	0.83		
AUC	0.85		

SVM, Support Vector Machine; AUC, area under the receiver operating characteristic curve.

## Data Availability

The data will be available upon reasonable request to the corresponding author.

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
