# Peer review of "Radiomics-Based Machine Learning for Predicting the Injury Time of Rib Fractures in Gemstone Spectral Imaging Scans"

_bioengineering, 2022, doi:10.3390/bioengineering10010008_

Round 1

Reviewer 1 Report

The paper presents the study of the prediction of the injury time of rib fractures in distinguishing fresh (30 days) or old (90 days) rib fractures. Authors propose the implementation of this prediction based on SVM. According to my experience and knowledge, I can evaluate this paper from the point of view of machine learning. But I can not evaluate the significance of the paper for the medical domain. Therefore my recommendations are for prediction aspects only.

  • Please explain the novelty and contribution of the study. Why this study is important and what academic values it adds to the field?
  • Please extend the analysis of the problem state of machine learning methods used in medicine. Why is SVM used? You use nonlarge sampling in the study, 111 instances are not large for classifier induction. Decision tree based classifiers and Bayesian classifiers are effective for such sampling, as a rule. Could you compare your investigation with the application of these classifiers? Or as minimal, be so kind as to justify the use of SVM only.
  • In the abstract the efficiency is evaluated by accuracy, but in the experimental study, precision is used for the evaluation (Table 2, Table 3). It is another metric for classifier evaluation. Could you introduce other metrics for the evaluation, for example, such as sensitivity, availability, F1, etc.?
  • You obtained the evaluation of the classifier as 0.85/0.81 only. It is not a very well result for the medical domain. How way can you propose for the result to improve? Could you consider the fuzzy classifier as one of the possible decisions for the future? For example, this classifier is considered in paper:
    • V. Levashenko, E. Zaitseva and S. Puuronen, "Fuzzy Classifier Based on Fuzzy Decision Tree," EUROCON 2007 - The International Conference on "Computer as a Tool", 2007, pp. 823-827, doi: 10.1109/EURCON.2007.4400614.
    • Martens, J., Wets, G., Vanthienen, J., Mues, C. An initial comparison of a fuzzy neural classifier and a decision tree based classifier, Expert Systems with Applications, 1998, 15(3-4), pp. 375-381

Author Response

Reviewer 1

The paper presents the study of the prediction of the injury time of rib fractures in distinguishing fresh (30 days) or old (90 days) rib fractures. Authors propose the implementation of this prediction based on SVM. According to my experience and knowledge, I can evaluate this paper from the point of view of machine learning. But I can not evaluate the significance of the paper for the medical domain. Therefore my recommendations are for prediction aspects only.

Comments: Please explain the novelty and contribution of the study. Why this study is important and what academic values it adds to the field?

Reply: Thank you for reviewing our work. As mentioned in the Introduction section, accurate detection of rib fractures with the injury time is expected to add value to the clinical practice. The knowledge of the injury time of rib fractures is expected to facilitate the distinction of artificial injuries in forensic medicine [1]. This was also reported by Talbot et al. [2], wherein the findings indicated that by recognizing and accurately reporting the imaging findings like the injury time of rib fractures will add value to the care given by radiologists to patients with thoracic trauma.

  1. Yang, C.; Zhao, J.; He, G.; et al. Deep learning into intelligent detection of rib fracture from x-ray imagery. Forensic Sci. Technol. 2021, 46, 134–139.

  1. Talbot, B.S.; Gange, C.P.; Chaturvedi, A.; Klionsky, N.; Hobbs, S.K.; Chaturvedi, A. Traumatic rib injury: Patterns, imaging pitfalls, complications, and treatment. RadioGraphics. 2017, 37, 628–651. DOI:10.1148/rg.2017160100.

Comments: Please extend the analysis of the problem state of machine learning methods used in medicine. Why is SVM used? You use nonlarge sampling in the study, 111 instances are not large for classifier induction. Decision tree based classifiers and Bayesian classifiers are effective for such sampling, as a rule. Could you compare your investigation with the application of these classifiers? Or as minimal, be so kind as to justify the use of SVM only.

Reply: We had expanded on the analysis of the problems addressed using machine learning methods in medicine in the Discussion under “Machine learning focused on how computers learn from data [22] could provide some additional medical information from imaging which is hard for human eyes to explain in medical practice; patients may benefit from this medical information based on machine learning approaches. Ji et al. used machine-learning analysis for contrast-enhanced CT radiomics to predict the recurrence of hepatocellular carcinoma after resection [23]; Draelos et al. used machine-learning models to predict multiple abnormalities with large-scale chest CT volumes [24].”

Kindly note that, for model construction, we have conducted experiments based on various machine learning models, including Gaussian Bayesian classifiers (GNB) and the LR classifier. Experimental results are provided in Supplementary Material 1. These suggest that their performances are inferior compared to SVM.

  1. Deo, R.C. Machine Learning in Medicine. Circulation 2015, 132, 1920-1930.
  2. Ji, G.W., Zhu, F. P., Xu, Q., Wang, K., Wu, M.Y., Tang, W.W., et al. Machine-learning analysis of con-trast-enhanced CT radiomics predicts recurrence of hepatocellular carcinoma after resection: A mul-ti-institutional study. Ebiomedicine 2019, 50, 156-165.
  3. Draelos, R.L., Dov, D., Mazurowski, M.A., Lo, J.Y., Henao, R., Rubin, G.D., et al. Machine-learning-based multiple abnormality prediction with large-scale chest computed tomography volumes. Med Image Anal 2021, 67, 101857.

Comments: In the abstract the efficiency is evaluated by accuracy, but in the experimental study, precision is used for the evaluation (Table 2, Table 3). It is another metric for classifier evaluation. Could you introduce other metrics for the evaluation, for example, such as sensitivity, availability, F1, etc.?

Reply: Kindly note that in Tables 2 and 3, we used precision, recall, F1 score, accuracy, and AUC as the metrics for classifier evaluation. In the Abstract, to succinctly present the experimental results, we only selected representative accuracy and AUC as the metrics to demonstrate model’s evaluation.

The sensitivity and specificity have the same meaning as the recall of positive and negative, so we did not repeat the calculation.

Comments: You obtained the evaluation of the classifier as 0.85/0.81 only. It is not a very well result for the medical domain. How way can you propose for the result to improve? Could you consider the fuzzy classifier as one of the possible decisions for the future? For example, this classifier is considered in paper:

  1. Levashenko, E. Zaitseva and S. Puuronen, "Fuzzy Classifier Based on Fuzzy Decision Tree," EUROCON 2007 - The International Conference on "Computer as a Tool", 2007, pp. 823-827, doi: 10.1109/EURCON.2007.4400614.

Martens, J., Wets, G., Vanthienen, J., Mues, C. An initial comparison of a fuzzy neural classifier and a decision tree based classifier, Expert Systems with Applications, 1998, 15(3-4), pp. 375-381

Reply: Thank you for the recommendation. Our follow-up research will take note of the fuzzy classifier to improve the model’s performance.

Reviewer 2 Report

Dear authors,

Thank you for submitting the manuscript bioengineering-2015803 for publication. The research derives from the field of radiomics in medical imaging and applies deep learning to predict fracture age using spectral CT imaging. Manual segmentation with ITK-SNAP is followed by Radcloud feature extraction and Lasso feature selection. A standard SVM Classifier is supported by human collaboration. The authors conclude that DL improves human prediction by ca. 10%.

The manuscript is written in sound structure and formal language, the aims meet the journal's interests, and the results are of immense importance for forensic documentation.

The reviewer's comments for some necessary improvements and clarification are listed below. A point-by-point response would be appreciated and would speed up the revision process.     

C001: N=111 patients were enrolled, and N=54 were included. Could you please highlight the exclusion criteria?

C002: L92-93. The definition of fracture age groups should be elaborated in detail. Kindly revise your results by describing the EXACT fracture age (mean-SD) based on the first diagnosis in each group of the training dataset. The description "...>30 days "and "... > 90 days" in too broad and might inject bias.   

C003: In line with C002, kindly describe how many "30 days" and how many "90 days" patients you included in the training group. Please include a STARD diagram to clarify issues C001 – C003.  

C004 – L29-30 The AUC analysis describes the accuracy of the classifier to detect >30 and >90 days fractures from a pool of fractures and controls. It is of immense importance to report the DISCRIMINATIVE CAPACITY of the classifier to DIFFERENTIATE the age of a given fracture between >30 and >90 days. Please revise your results accordingly and update this information in the abstract.  

C005: L29-30. In line with C004, in the abstract, it is stated: "…to differentiate fracture age…". This is not correct. The ROC analysis describes the detection accuracy but does not provide information about the differentiation. Kindly address this issue.   

C006 – The gold standard is the radiological diagnosis. According to (L98), could you elaborate on the reason of excluding the radiologists from the study?

C007: Please improve the figure quality. Figure 3 and Fig 4 are very condensed; to improve readability, they should be split and/or magnified. For example, the confusion matrix could have smaller boxes with larger font.

C008: provide sample pictures of Ca- and iodine decomposition images for fractures >30 and  > 90 days, as well as from fresh fractures, to better visualize your aims.

C009: The reviewer does not understand the necessity for (expensive !! ) spectral imaging for this study. Even if patients were investigated with the use of iodine contrast enhancer, the amount of enhancer in the fracture area is amenable. Since the spectral imaging is part of the MS-title, the authors should JUSTIFY the GSI NECESSITY over the MONOENERGETIC IMAGING at THE FIRST PLACE.  My primary motivation for this comment is to increase the manuscript's citability by promoting the method for monoenergetic scanners, as well (cheaper and more broadly available).

Author Response

Reviewer 2

Thank you for submitting the manuscript bioengineering-2015803 for publication. The research derives from the field of radiomics in medical imaging and applies deep learning to predict fracture age using spectral CT imaging. Manual segmentation with ITK-SNAP is followed by Radcloud feature extraction and Lasso feature selection. A standard SVM Classifier is supported by human collaboration. The authors conclude that DL improves human prediction by ca. 10%.

The manuscript is written in sound structure and formal language, the aims meet the journal's interests, and the results are of immense importance for forensic documentation.

The reviewer's comments for some necessary improvements and clarification are listed below. A point-by-point response would be appreciated and would speed up the revision process.    

C001: N=111 patients were enrolled, and N=54 were included. Could you please highlight the exclusion criteria?

Reply: Thank you for reviewing our work. The exclusion criterion was as follows: the data files of GSI scans for image postprocessing were missing.

C002: L92-93. The definition of fracture age groups should be elaborated in detail. Kindly revise your results by describing the EXACT fracture age (mean-SD) based on the first diagnosis in each group of the training dataset. The description "...>30 days "and "... > 90 days" in too broad and might inject bias.  

Reply: In accordance with your pertinent suggestions, we have added the necessary information to our revised draft. The detailed corrections are listed in the STARD diagram as shown below.

C003: In line with C002, kindly describe how many "30 days" and how many "90 days" patients you included in the training group. Please include a STARD diagram to clarify issues C001 – C003. 

Reply: Accordingly, we have added the STARD diagram to address comments C001 – C003.

C004 – L29-30 The AUC analysis describes the accuracy of the classifier to detect >30 and >90 days fractures from a pool of fractures and controls. It is of immense importance to report the DISCRIMINATIVE CAPACITY of the classifier to DIFFERENTIATE the age of a given fracture between >30 and >90 days. Please revise your results accordingly and update this information in the abstract. 

Reply: Thank you for reviewing our work. Please be informed that we strongly agreed with your great suggestions, we will add this part into our future study. We did not join the part of the injury time of rib fractures between >30 and >90 days into the present study, because the study methods will be changed if we try to report a three-way classification.

C005: L29-30. In line with C004, in the abstract, it is stated: "…to differentiate fracture age…". This is not correct. The ROC analysis describes the detection accuracy but does not provide information about the differentiation. Kindly address this issue.  

Reply: Accordingly, we have corrected ‘differentiating’ to ‘differentiating’.

C006 – The gold standard is the radiological diagnosis. According to (L98), could you elaborate on the reason of excluding the radiologists from the study?

Reply: Kindly note that the gold standard is not a radiological diagnosis. The gold standard of injury time is the medical history from the medical records stored in our picture archives and communication system. The medical history of each patient was recorded as the accurate injury date before they underwent CT scanning.

C007: Please improve the figure quality. Figure 3 and Fig 4 are very condensed; to improve readability, they should be split and/or magnified. For example, the confusion matrix could have smaller boxes with larger font.

Reply: Accordingly, we have magnified Figures 3 and 4.

C008: provide sample pictures of Ca- and iodine decomposition images for fractures >30 and > 90 days, as well as from fresh fractures, to better visualize your aims.

Reply: Accordingly, we added a new image for better visualization of our aims as per your suggestion.

Figure 5. The calcium-based material decomposition images. Figure A shows the injury time of a rib fracture within 30 days; figure B shows the injury time of a rib fracture between 30 and 90 days, and figure C shows the injury time of a rib fracture over 90 days.

C009: The reviewer does not understand the necessity for (expensive !! ) spectral imaging for this study. Even if patients were investigated with the use of iodine contrast enhancer, the amount of enhancer in the fracture area is amenable. Since the spectral imaging is part of the MS-title, the authors should JUSTIFY the GSI NECESSITY over the MONOENERGETIC IMAGING at THE FIRST PLACE. My primary motivation for this comment is to increase the manuscript's citability by promoting the method for monoenergetic scanners, as well (cheaper and more broadly available).

Reply: We investigated the GSI imaging in this retrospective study due to the following reasons: there are no studies reporting the injury time of rib fractures, and we attempted to check whether the technique using imaging with radiomics could achieve this aim. There are no studies reporting which kind of CT imaging modality would add additional value to this aim. When we learned that the calcium-based material decomposition images from GSI could be used to measure the bone mineral content, we searched the GSI scans of patients with rib fractures in our hospital and attempted to investigate whether radiomics analysis could extract the segmented volume of interest (VOI) of each broken end of rib fractures to predict the possible injury time of ribs to distinguish the injury time of rib fractures within 30 days (approximately 4 weeks) or 90 days (approximately 12 weeks). We strongly agree with your comments, and we plan to show a comparison of results between GSI and MONOENERGETIC IMAGING modalities in future studies.

Reviewer 3 Report

The manuscript “Radiomics-based machine learning for predicting the injury 2 time of rib fractures in Gemstone Spectral Imaging scans” reports an interesting retrospective study on to predict the injury time of rib fracture in distinguishing fresh (30 days) or old (90 days) rib fractures using gemstone spectral imaging (GSI) through radiomics based machine learning. The study has been done systematically, and results are interesting for readers of Bioengineering. However, the discussion part is shallow and is not based on the results and its interpretation. I would request for rewriting of dissection based on results, avoiding general statements.   

Author Response

Reviewer 3

The manuscript “Radiomics-based machine learning for predicting the injury 2 time of rib fractures in Gemstone Spectral Imaging scans” reports an interesting retrospective study on to predict the injury time of rib fracture in distinguishing fresh (30 days) or old (90 days) rib fractures using gemstone spectral imaging (GSI) through radiomics based machine learning. The study has been done systematically, and results are interesting for readers of Bioengineering. However, the discussion part is shallow and is not based on the results and its interpretation. I would request for rewriting of dissection based on results, avoiding general statements. 

Reply: Please be informed that we had revised the discussion as follows:

“The present study aimed to predict the injury time of rib fractures based on radiomic features derived from the segmented VOI of each broken end of rib fractures in GSI scans. Our proposed SVM model showed a better prediction of the injury time of rib fractures by distinguishing 30 days or 90 days. The model displayed better performance than the experienced radiologist’s interpretation for both time points, thus indicating that the radiomics-based model could assist radiologists in evaluating the injury time of rib fractures with satisfactory performance.

Artificial intelligence techniques, including deep learning, radiomics, and radiomics-based machine learning, are valuable for clinical diagnosis and treatment in medical practice[14-19]. The majority of previous studies have focused on the detection and segmentation of rib fractures using deep learning[17,20]; nonetheless, few studies have attempted to classify rib fractures[20-21] and demonstrated that artificial intelligence could significantly reduce the workload of doctors while improving diagnostic accuracy. Machine learning focused on how computers learn from data [22] could provide some additional medical information from imaging which is hard for human eyes to explain in medical practice; patients may benefit from this medical information based on machine learning approaches. Ji et al. used machine-learning analysis for contrast-enhanced CT radiomics to predict the recurrence of hepatocellular carcinoma after resection [23]; Draelos et al. used machine-learning models to predict multiple abnormalities with large-scale chest CT volumes [24]. However, no reports have confirmed the injury time of rib fractures using such machine-learning techniques.

Rib fracture healing is affected by age[1-2] and evaluating the injury time of rib fractures from the imaging features of rib fractures is difficult even for an experienced radiologist (Fig.5), despite the existing knowledge (including morphological features) and experience. Our findings indicated and confirmed this phenomenon. First, the selected nine features (Table 1) correlated significantly with the grayscale. Machine learning showed better performance in distinguishing 30 days or 90 days (accuracy: 0.85 and 0.81, respectively) compared to radiologists’ interpretation (0.74 and 0.71, respectively). Our results indicated machine learning could learn some features to help predict the injury time of rib fractures. Moreover, our results showed that the better performance in distinguishing 30 days than the 90 days in both machine learning and radiologist’s interpretation, this may be explained by the natural characteristic of fracture healing. The less confusing imaging features of fracture healing within 30 days may be easy to predict the injury time. However, an unsupervised radiomics-based model may not be superior to human performance despite the higher accuracy. Radiologists’ interpretation can be assisted by the prediction performance of the model; the increased accuracy of distinguishing 30 days and 90 days displayed effective complementarity following human-model collaboration, consistent with the results of previous artificial intelligence-based studies[17, 19]. In the present study, we set the injury time of fresh fractures within 4 weeks (30 days). This could be attributed to the absence of an accepted clinical definition of fresh fractures[4] and the influence of age. Moreover, we set the injury time at 12 weeks (90 days) owing to the consensus of old fractures. Following rib fracture at 30 days, the imaging features principally comprised a sharp fracture line without periosteal reaction or callus formation, which was easily captured by the model or detected by human eyes. However, a comparison of the features up to 90 days, like blurred edges of the fracture line with callus formation, mature callus, bone remodeling, and invisible features of the fracture line, added to the confounding factors for the evaluation. This finding may explain the better prediction of the injury time of rib fractures within 30 days. In the present study, we summarized the performance of other machine learning models (RF, LR, and GNB) in Supplementary File 1. Some models showed better performance than SVM in terms of overfitting; some models did not show satisfactory performance compared to SVM, and finally, we chose SVM as the best model.

Figure 5. The calcium-based material decomposition images. Figure A shows the injury time of a rib fracture within 30 days; figure B shows the injury time of a rib fracture between 30 and 90 days, and figure C shows the injury time of a rib fracture over 90 days.

The present study had several limitations. Researchers have established GSI scans as a novel tool to provide additional information for clinical decision-making [7-11]; thus, we used GSI data for rib fracture segmentation. However, we did not compare the performance of radiomics prediction between the conventional CT images and GSI scans with different image modalities; our findings could not indicate whether GSI scans could contribute more than conventional CT images, and this needs to be addressed in future studies. Nevertheless, our radiomics prediction already demonstrated better performance both by model and human-model collaboration. Second, we performed a single-center study without external validation datasets; thus, determining the model’s robustness was challenging, and will be addressed in future studies.

In conclusion, the imaging features derived from the GSI scan-based radiomics model displayed good accuracy in differentiating the injury time of rib fractures within 30 and 90 days, and the human-model collaboration demonstrated more accurate outcomes. Our novel study used a radiomics-based model to predict the injury time of rib fractures and demonstrated better performance, and is expected to be valuable in clinical and forensic decision-making.”

Round 2

Reviewer 1 Report

As I noted in the previous review, the authors consider an interesting and relevant problem. However, the proposed version of the manuscript (after correction) still does not show  ​​the novelty and significance of the result. The authors pay much attention to the trivial task of dividing the sample into trainable and test parts. But they do not show the specifics of their study.

As before, I recommend to:

- extend the analysis of medical imaging research (classification, diagnosis predictions,  methods analysis, and comparison);

- more specifically indicate the novelty of the result;

- explain how the Human-model collaboration is built in more detail;

- explain the use of SVM (comment " Some models showed better performance than SVM in terms of overfitting; some models did not show satisfactory performance compared to SVM, and finally, we chose SVM as the best model " - unacceptable for scientific publication);

- explain the acceptability of the obtained experimental studies. The obtained classification results (predictions) are rather weak. In most studies, accuracy/precision is achieved at the level of 0.95 and higher. Is it the specifics of the initial data or of the problem?

- explain the choice of metrics for evaluating the result. Why choose precision and F1-score?

- consider future studies which can improve the presented result

Author Response

Comments and Suggestions for Authors

As I noted in the previous review, the authors consider an interesting and relevant problem. However, the proposed version of the manuscript (after correction) still does not show ​​the novelty and significance of the result. The authors pay much attention to the trivial task of dividing the sample into trainable and test parts. But they do not show the specifics of their study.

As before, I recommend to:

- extend the analysis of medical imaging research (classification, diagnosis predictions, methods analysis, and comparison);

Reply: Thank you for reviewing our work again. Your comments suggested extending the analysis of medical imaging research, we had added the analysis of former studies using radiomics methods in medical imaging research according to previous revised version of the manuscript. If your comments suggested extending the analysis of medical imaging research of our results, please be informed that we hypothesized that radiomics analysis could extract the segmented volume of interest (VOI) of each broken end of rib fractures in GSI scans to predict the possible injury time of ribs, and this had not been reported in previous studies. In the present study, the results showed the feasibility of this method, we will use this method in future study of clinical research. So, if we did not understand your comments, we are looking forward to hear from you.

- more specifically indicate the novelty of the result;

Reply: Thank you for reviewing our work again. As our study has not been reported for predicting the injury time of ribs, we thought this was the novelty of this study, and our results supported this hypothesis. If this hypothesis is true, it can be applied to clinical research in the future. So, we are looking forward to hear from you that what more specifical results you want us to indicate.

- explain how the Human-model collaboration is built in more detail;

Reply: Thank you for reviewing our work again. Please be informed that human-model collaboration was built through the following steps: The model predicted the injury time independently, then the radiologist interpreted the injury time with the assisted results from the model, the radiologist could modify the results of model if he/she did not approve the prediction. A chest radiologist with 12 years of experience in rib fracture diagnosis was invited in this study. The combination of model and radiologists were regarded as the final human-model collaboration results.

- explain the use of SVM (comment " Some models showed better performance than SVM in terms of overfitting; some models did not show satisfactory performance compared to SVM, and finally, we chose SVM as the best model " - unacceptable for scientific publication);

Reply: Thank you for your valuable suggestions. In section 3.3.1, we added the reason for selecting SVM. The details are as follows: “Compared with Decision tree and Bayesian inference based classifier, SVM is less susceptible to data distribution, thus ensuring the stability of the model. Compared with other property line classifiers, when faced with linear indivisibility, SVM makes the samples in low dimension input space linearly separable by adding relaxation variables and using nonlinear mapping to map them to high dimension space, so that it can find the optimal classification hyperplane in the feature space. Therefore, we finally choose SVM as the classifier.”

- explain the acceptability of the obtained experimental studies. The obtained classification results (predictions) are rather weak. In most studies, accuracy/precision is achieved at the level of 0.95 and higher. Is it the specifics of the initial data or of the problem?

Reply: Thank you for your nice comments on our article. Our model obtained 0.82 and 0.81 positive precision in the testing set at 30 day labels and the testing set at 90 day labels respectively. This may be due to the small amount of data and the large within-class distance. For example, in the case of fracture greater than 90 days, mean-SD is 306.5-660.25. The large within-class distance makes it difficult to summarize the common features of the same class when classifying models, which affects the performance of models. In some exploratory studies [1-3], the accuracy/precision of the model may not reach a high level, but it provides a new research method.

[1] Basler, L. , et al. "Radiomics, Tumor Volume, and Blood Biomarkers for Early Prediction of Pseudoprogression in Patients with Metastatic Melanoma Treated with Immune Checkpoint Inhibition." Clinical cancer research : an official journal of the American Association for Cancer Research 26.16:4414-4425.2020

[2] Dercle M L ,  Lu L ,  Schwartz L H , et al. Radiomics Response Signature for Identification of Metastatic Colorectal Cancer Sensitive to Therapies Targeting EGFR Pathway[J]. JNCI: Journal of the National Cancer Institute, 2020.

[3] Liu J ,  Sun D ,  Chen L , et al. Radiomics Analysis of Dynamic Contrast-Enhanced Magnetic Resonance Imaging for the Prediction of Sentinel Lymph Node Metastasis in Breast Cancer[J]. Frontiers in Oncology, 2019, 9:980-.

- explain the choice of metrics for evaluating the result. Why choose precision and F1-score?

Reply: Thank you for your nice comments on our article. In Table 2 and Table 3, we used precision, recall, F1 score, accuracy, and AUC as metrics for classifier evaluation. Among them, there is an inverse relationship between precision, recall, F1 score of one category and precision, recall, F1 score of another category. Excessively pursuing high precision, recall, F1 score of one category means reducing precision, recall, F1 score of another category. Therefore, we finally choose AUC as a comprehensive metric to evaluate the model.

- consider future studies which can improve the presented result

Reply: Thank you for your valuable suggestions. We will increase the number of training sets to improve the learning ability of the model and achieve better results. At the same time, we will also balance the sample size of different categories to avoid the influence of unbalanced samples on the model results.

Reviewer 2 Report

Dear authors, 

thanks for providing your manuscript corrections. From the reviewer’aspect   all issues have been tackled. I wish you a successful publication. 

Author Response

Thank you for reviewing our work again.

Reviewer 3 Report

This revised version of the manuscript responses all my queries satisfactorily. I would recommend for publication. 

Author Response

Thank you for reviewing our work again.